# Structures of HCoV-OC43 HR1 Domain in Complex with Cognate HR2 or Analogue EK1 Peptide

**DOI:** 10.3390/v17030343

**Published:** 2025-02-28

**Authors:** Xiuxiu He, Huanzhen Liu, Guang Yang, Lei Yan

**Affiliations:** 1School of Food Science and Pharmaceutical Engineering, Nanjing Normal University, Nanjing 210023, Chinayangguang@shanghaitech.edu.cn (G.Y.); 2Shanghai Institute for Advanced Immunochemical Studies, ShanghaiTech University, Shanghai 201210, China; 3International Research Center of Synthetic Biology, Nanjing Normal University, Nanjing 210023, China

**Keywords:** HCoV-OC43, fusion core, heptad repeats, fusion inhibitor EK1, viral entry

## Abstract

Human coronavirus OC43 (HCoV-OC43) is usually associated with common colds, but also related to severe disease in the frail. Its envelope glycoproteins spike (S) is responsible for host-cell attachment and membrane fusion. To understand the molecular basis of membrane fusion of HCoV-OC43, we solved the 3.34 Å crystal structure of the post-fusion state formed by two heptad repeat domains (HR1P and HR2P) of OC43-S. This fusion core comprises a parallel trimeric coiled coil of three HR1 helices with 61 Å at length, around which three HR2 helices are entwined in an antiparallel manner, as anticipated. Moreover, a pan-CoV fusion inhibitor EK1 derived from OC43-HR2P was also crystalized with OC43-HR1P in the resolution of 2.71 Å. Parallel comparisons rationalize the design of EK1, maintaining various hydrophobic and charged or hydrophilic interactions formed in the initial fusion core to stabilize the overall conformation. Together, our results not only reveal the critical intrahelical and interhelical interactions underlying the mechanism of action of OC43-S fusion, but also help our understanding on the mechanism of HCoV-OC43 inhibition by analogue HR2 mimic peptide.

## 1. Introduction

Coronaviruses (CoVs) are known as large, positive-sense RNA enveloped viruses in the *Nidovirales* order and can be grouped into four genera: *α*, *β*, *γ* and *δ* genus [1]. To date, seven CoVs have been identified with capacity of infecting humans, including *α*-CoV: HCoV-229E and HCoV-NL63, *β*-CoV: HCoV-HKU1 and HCoV-OC43, and three highly pathogenic CoVs–severe acute respiratory syndrome coronavirus (SARS-CoV or SARS-CoV-1), Middle East respiratory syndrome coronavirus (MERS-CoV) and the pandemic SARS-CoV-2 [2]. Amongst, human coronavirus (HCoV) OC43 is a globally circulating virus sustained by recurrent reinfections seasonally. It is usually endemic in human population, generally causing mild common cold infections, occasionally bringing out severe complications or fatalities in young children, the elderly and immuno-compromised individuals due to severe lower respiratory tract illness, including bronchitis and pneumonia [3,4]. As yet, no vaccines or specific antiviral treatments are available for HCoV-OC43 prophylaxis or treatment in clinical practice.

CoVs use the transmembrane homo-trimeric glycoprotein spike (S) to attach to host receptors and fuse the viral and cellular membranes for entry of genetic materials [5,6]. Similar to other CoVs, HCoV-OC43 is also assigned to class I enveloped virus family [7]. Shortly, in the native state, HCoV-OC43 S protein presents as an inactive precursor. During the viral infection process, it is supposed that proteases on target cell membrane can activate the S protein by cleaving it into S1 and S2 subunits [8]. The S1 subunit contains N-terminal domain (NTD) for binding 9-O-acetylated sialic acid, whereas the C-terminal domain (CTD) has an unknown function [8,9]. The S2 subunit, which includes fusion peptide (FP), heptad repeat 1 (HR1), heptad repeat 2 (HR2), transmembrane domain (TM) and cytoplasmic domain fusion (CP), mediates viral fusion and entry [10]. S2 undergoes dramatic conformational changes after receptor binding in S1 domain and cleavage, including insertion of FP into the target cell membrane, exposure of the pre-hairpin coiled coil of HR1 domain, and the interaction between viral HR2 domain and HR1 trimer to form six-helix bundle (6-HB), thus bringing the viral envelope and cell membrane into close proximity for viral fusion and entry [8,11,12].

Here, we solved the crystal structure of fusion core of HCoV-OC43 composed by HR1 and HR2 derived peptides, at 3.34 Å resolution. The three HR1 helices form a central helix through the hydrophobic interaction. HR2 then binds to the side groove formed by the 3-HR1 coiled coil. We illustrate that there is a contact network between HR1 and HR2 domains, involving extensive hydrophobic and hydrogen bonding interactions. In addition, the hydrophobic grooves of the 3HR1 coils are complementary to accommodate the spiral half and extension half of HR2, respectively. At the same time, previously, our collaborators have demonstrated peptide EK1 derived from OC43-HR2 sequence could broadly bind to the homo-trimeric HR1s [13]. It is putative that EK1 interacts with the viral exposed HR1-trimer and competitively inhibit viral autologous 6-HB formation, thus blocking virus–cell membrane fusion. Therefore, it is essential to solve the coiled-coil structure and characterize the interaction inter and inner HR1 and HR2 complex for better understanding the entry mechanism of HCoV-OC43 and designing of novel peptide-based fusion inhibitor.

## 2. Materials and Methods

### 2.1. Peptide Synthesis

Peptides of OC43-HR1P (Ac-ANAFNNALDAIQEGFDATNSALVKIQAVVNANAE ALNNLLQQ-NH2), OC43-HR2P (Ac-SLDYINVTFLDLQDEMNRLQEAIKVLNQSYINL KDIG-NH2) and EK1 (Ac-SLDQINVTFLDLEYEMKKLEEAIKKLEESYIDLKELG-NH2) were synthesized by solid-phase peptide synthesis at GenScript Inc. Company (Nanjing, China) and validated by LC-MS.

### 2.2. Native Polyacrylamide Gel Electrophoresis (Native-PAGE)

OC43-HR1P, OC43-HR2P and their mixture were simultaneously incubated at 25 °C for 30 min, followed by segregation by 18% Tris-glycine gel with constant 120 V at room temperature for 2 h. The gel was stained with staining solution with Coomassie bright blue dye (Beyotime Biotechnology, Shanghai, China) and imaged with a Tanon 2500-B scanner (Tanon Science and Technology, Shanghai, China). Gray values were extracted by ImageJ V1.8.0.112 and the curve was drawn by GraphPad Prism 10.

### 2.3. Circular Dichroism (CD) Spectroscopy

The secondary structure of peptides OC43-HR1P and OC43-HR2P of equal amounts mixture was determined and characterized by CD spectroscopy method. In brief, the OC43-HR1P/HR2P complex were diluted in phosphate-buffered saline (PBS) (pH = 7.2) to 50 µM and balanced and incubated at 37 °C for 30 min. CD spectra were acquired on a Chirascan-plus Circular Dichroism Spectrometers (Applied Photophysics, Leatherhead, UK) from 195 to 260 nm, using the optical path length of 0.1 cm and bandwidth of 1 nm at room temperature. The baseline was determined using PBS. The wavelength at 222 nm was used to monitor α-helix content, with gradual increase in the temperature. Measurements at small temperature intervals was performed to obtain a precise melting curve during a temperature ramp of 1 °C/min from 15–75 °C.

### 2.4. Assembly of OC43-HR1P in Complex with OC43-HR2P or EK1

OC43-HR2P and EK1 were dissolved in the buffer containing 40 mM TRIS-HCl pH 8.0 and 300 mM NaCl, while OC43-HR1P peptide was dissolved in pure deionized water and pH was adjusted until clear and transparent with 10 mM NaOH solution to make a stock at 2.25 mM concentration. Equimolar amounts of OC43-HR1P and OC43-HR2P or EK1 peptide were mixed to reach a concentration of 1.125 mM. OC43-HR1P trimer and OC43-HR1P/HR2P complex or OC43 HR1-EK1 complex were all incubated at the room temperature for at least 3 h. All peptide stocks were stored at −80 °C for different uses.

### 2.5. Protein Crystallization

Initial protein crystallization experiments were screened against 384 individual crystallization conditions from Hampton Research with sitting-drop vapor diffusion method. Finally, diffraction-quality crystals grew at 16 °C for about one week via the hanging drop vapor diffusion method by mixing equal volume of protein solution [HCoV-OC43 6-HB: 10 mg/mL] and reservoir solution, [0.2 M ZnAc_2_, 0.1 M Sodium cacodylate pH 6.5 and 18% PEG 8000]. Crystals of HR1(OC43)-EK1 were obtained by similar means in mother liquid containing 0.2 M MgCl_2_, 0.1 M TRIS-HCl pH 8.5 and 30% PEG 4000. Then, crystals were collected and flash-frozen in liquid nitrogen for about 10 s, after immersing in mother liquid supplemented with 10% glycerol for cryoprotection. All the crystals were kept in liquid nitrogen until diffraction.

### 2.6. Data Collection, Structure Determination, and Refinement

Crystals of HR1(OC43)-HR2 and HR1(OC43)-EK1 were diffracted at beamline BL02U1 (previously known as BL17U1) and BL19U1 of the Shanghai Synchrotron Radiation Facility (SSRF), respectively, under the condition of 77 K during data collection. Then, data were indexed and scaled with HKL3000 v723 [14]. Phases were solved by the molecular replacement method using PHENIX phaser v1.16-3549 [15]. All refinement procedures were carried out by alternating between automatic PHENIX refine v1.16-3549 [15] and manual WinCoot v0.9.8.95 [16], round by round. Detailed statistics of data collection and refinement are shown in Table 1 as below.

### 2.7. Structure Presentation

All of the structures were drawn through the PyMOL software 3.1 [17]. Superimposition was conducted in WinCoot with LSQ superimpose program. The electron density document used in PyMOL was generated by CCP4 with FFT program [18]. The electrostatic surface was plotted by APBS and PDB2PQR plugin embedded in PyMOL [19].

## 3. Results

### 3.1. Design and Biophysical Characterization of OC43-HR1P and OC43-HR2P

The S2 protein of HCoV-OC43 contains a N-terminal fusion peptide (FP, residues 904–943), a N-terminal heptad repeats 1 domain (HR1, residues 1004–1054), a C-terminal heptad repeats 2 domain (HR2, residues 1248–1286), a transmembrane domain (residues 1298–1318), and an intracellular domain (residues 1319–1353) (Figure 1A). We herein report the design and synthesis of two peptides, designated OC43-HR1P and OC43-HR2P, spanning residues 1008–1049 in the HR1 domain and 1252–1288 in the HR2 domain of HCoV-OC43 spike protein, respectively (Figure 1A). To investigate whether OC43-HR1P and OC43-HR2P interact to form a six helical bundle (6-HB), we incubated OC43-HR1P in phosphate-buffered saline (PBS) at 40 µM with OC43-HR2P at gradient 10, 20, 40, and 80 µM, respectively, at 37 °C for 30 min before loading the samples to the Tris-glycine gel, followed by analysis with Native-PAGE as previously described. As shown in Figure 1B, OC43-HR1P alone displayed no band in the gel (lane 8) because this peptide carries net positive charges, thus moving up and off the gel under the native electrophoresis condition. OC43-HR2P alone showed a band at the lower part in the gel (lane 7). However, the mixtures of OC43-HR2P with OC43-HR1P gradient concentrations, showed new bands corresponding to the 6-HB at the upper part in the gel (lanes 1–6), confirming that OC43-HR1P and OC43-HR2P do interact with each other to form 6-HB in a dose-dependent manner with a kD value of around 28.53 µM (Figure 1B). To further study characteristics of the interaction between OC43-HR1P and OC43-HR2P, we determined the secondary structures of OC43-HR1P and OC43-HR2P and their complex in mixture (OC43-HR1P/HR2P) before and after the formation of fusion core by circular-dichroism (CD) spectroscopy. In solution, as shown in Figure 1C, OC43-HR1P/HR2P were mixed at equimolar concentration together to form 6-HB, showcasing a complex with high α-helicity, as characterized by the saddle-shaped negative peak at 208 and 222 nm in the far UV region of the CD spectrum. Meanwhile, this helical bundle showed thermal stability with a Tm of 73.5 ± 0.65 °C (Figure 1D). The above results confirm that synthesized peptides derived from HCoV-OC43 HR1 and HR2 domains harbored the interacting capacity to be used for structural study.

### 3.2. Overall Structure of HCoV-OC43 Fusion Core

OC43-HR1P and OC43-HR2P were mixed at equal amounts and run on the Superdex 75 pg column by gel filtration after incubation at room temperature. The peak indicating the complex turned out to elute at about 70 mL corresponding to 20 kD molecular weight (Figure 2A). Finally, the OC43-HR1P/HR2P complex crystallized in space group P222_1_, with unit-cell parameters of a = 39.0 Å, b = 48.2 Å and c = 103.3 Å and three molecules per asymmetric unit (Table 1). The structure was solved by molecular replacement using the crystal structure of the MHV fusion core (PDB entry 1WDG) as the search model, and was refined to a final resolution of 3.34 Å with an R_work_ of 24.0% and an R_free_ of 28.1% (Table 1). OC43-HR1P displays a 11-turn-helix, while OC43-HR2P adopts a mixed conformation: residues 1263–1280 fold into a five-turn helix, while residues 1256–1263 and 1281–1282 on either side of the helix adopt an extended conformation (Figure 2B).

The overall structure of the HR1-HR2 domain of HCoV-OC43 represented a canonical 6-HB structure (Figure 2C). Taking a rod-like shape with a length of ~61 Å and a diameter of ~18 Å, HCoV-OC43 fusion core contains a parallel trimeric coiled-coil structure of three HR1 helices (residues 1008–1048, gray), around which three HR2 helices (residues 1256–1283, cyan) are entwined in an antiparallel manner (Figure 2C). According to the structure shown in electrostatic potential surface, the HR2 region of HCoV-OC43 can be divided into two parts: linear N- and C-terminal regions flanking a helical region laying in the deep groove (Figure 2D). In general, the fusion core from HCoV-OC43 and MHV adopt a similar fold, consistent with their high sequence identity and similarity between the two proteins. Comparing the fusion core structure of MHV (PDB entry 1WDG) [20] with that of HCoV-OC43, we found that their HR1 and HR2 domains overlap with each other very well with a root mean square deviation of 0.93 Å for all Cα atoms of one HR1-HR2 protomer (Figure 2E).

### 3.3. Interactions Inner HCoV-OC43 6-HB Fusion Core

The three HR1 helices of HCoV-OC43 are closely packed against each other by hydrophobic force in a parallel manner. The hydrophobic packing forms by residues in the position “d” (F1001, I1018, I1032 and L1046, respectively) and “ga” (^1014^AL^1015^, ^1021^GF^1022^, ^1028^AL^1029^, ^1035^VV^1036^ and ^1042^AL^1043^, respectively) (Figure 3A), whileT1025 and N1039 in position “d”, although polar in nature, are positioned in a way that their side chains participate in the local hydrophobic packing through interactions with neighboring hydrophobic residues. No stutter was observed in the arrangement of heptad repeats. The buried area for each HR1P reaches 1038 Å^2^. Three HR2 helices interact with HR1 helices mainly through hydrophobic interaction between hydrophobic residues in HR2 regions and the grooves on the surface of the central coiled coil (Figure 3B). HR2P peptides are well packed against the hydrophobic grooves of a central three-helical coiled coil. Similar to those fusion cores, HR2 helices of HCoV-OC43 also contain OXO motifs, in which O represents a hydrophobic residue, and X represents any residue [21]. ^1256^INV^1258^ and ^1261^LDL^1263^ in HR2 regions are both composed of OXO motifs; the side chains of the O residues inset into or align with the hydrophobic grooves of the central coiled coil, whereas the side chains of the X residues are directed into solvent (Figure 3B right panel). The OXO motifs are responsible for the partially extended conformation of HR2, and this pattern also makes the fusion core stable in solvent, as most of the hydrophobic residues in HR2 helices are packed against the central coiled coil, leaving the hydrophilic residues exposed to solvent. Moreover, at the junction region between linear and helical parts of HR2P, a strong hydrogen bond network is noted. For hydrogen bonds, the amide group of side chain of Q1033 in HR2 interacts with the side chain of Q1264 and the backbone of L1263 in one HR1 helix, while the side chain of N1037 interacts with backbones of L1261 and D1262 in HR1 helix (Figure 3C). These relatively concentrated hydrophilic interactions constitute an anchoring point in the middle of HR2 domain, stabilizing the linear and helical region of HR2P and also the whole 6-HB conformation. Electrostatic stapling between HR1P and HR2P is observed by salt bridge formed between R1269 and E1020 as well as D1023 and partially between D1262 and K1031 (Figure 3C).

### 3.4. Molecular Mechanism Underlying Formation of HCoV-OC43 HR1-EK1

EK1 peptide is an antiviral peptide designed primarily to combat coronavirus infections, including SARS-CoV and MERS-CoV. It targets the viral spike (S) protein, interfering with the fusion process between the virus and host cell membranes, thereby inhibiting viral entry. EK1 has demonstrated broad-spectrum anti-coronavirus activity in multiple studies and proven its efficacy in animal models. As a potential antiviral therapy, EK1 peptide may offer new strategies for the prevention and treatment of coronavirus infections [10,15,16]. We revisited the design of EK1 derived from OC43-HR2P by elucidating the interactions between OC43-HR1P and EK1, as observed in the crystal structure. The HR1(OC43)-EK1 complex crystallized in space group P3, with cell parameters a = b = 53.0 Å, c = 62.1 Å and angles of 90°, 90° and 120° (Table 1). The structure was solved by molecular replacement, using the crystal structure of the HCoV-OC43 fusion core (PDB entry: 8ZX7) as the search model and refined to a final resolution of 2.79 Å with R_work_ of 28.2% and R_free_ of 31.3% (Table 1). HR1P (residues 1008–1049) of HCoV-OC43 forms an 11-turn α-helix, while EK1 adopts mixed conformations: residues L12-S29 fold into a five-turn α-helix while residues I5-D11 and Y30-I31 on either side of the helix adopt an extended conformation (Figure 4B). HR1P from three HR1(OC43)-EK1 molecules are arranged around the crystallographic three-fold symmetry axis and form the central hydrophobic core. Then, three EK1 helices pack in an oblique, left-handed and antiparallel direction into hydrophobic grooves on the surface of this trimeric coiled coil, forming in a six-helical bundle structure with dimensions of ~62 Å in length and 19 Å in diameter with a left-handed supercoil (Figure 4B and Figure 5C). The central region of EK1 folds into a five-turn helix, which packs against two neighboring HR1 helices via extensive hydrophobic interactions and a few electrostatic or polar interactions (Figure 4D,E). 3HR1 cores of HCoV-OC43 are illustrated as grey surfaces, residues on EK1 that are involved in hydrophobic packing are shown as stick models. In detail, residues L12, E15, M16, L19, A22, I23, L26, S29, and Y30 on EK1 are located within the five-turn helix of EK1 and form strong hydrophobic interactions with the 3HR1 cores (Figure 4E, left and middle panel). Moreover, residues E13, E15, K18, E20, E27, E28, and Y30 are located within the five-turn helix region of EK1 and interact with HR1 residues through side chain-to-side chain hydrophilic interaction. In the extended region of EK1, polar interactions between side chain (HR1 residues) and main chain (EK1 residues) dominate. Most of these side chain-to-main chain polar interactions cluster at either end of the EK1 helical region, which intertwine into extensive H-bond networks and likely help secure the EK1 helical region in the correct register (Figure 4E, right panel). Hydrophobic residues in the extended region of EK1 also insert their bulky side chains into hydrophobic pockets on the surface of the 3HR1 cores (residues I5, V7, L10 and I31), further strengthening the adhesion of the EK1 extended region onto the 3HR1 cores (Figure 4E, middle panel).

In summary, the EK1 peptide is proposed to outcompetes self-derived HR2 of HCoV-OC43, hence inhibiting completement of fusion (Figure 5A).

## 4. Discussion

Receptor recognition and membrane fusion are two crucial steps for entry of enveloped viruses [28]. The molecular basis of receptor binding by HCoV-OC43 and antigenic definition of corresponding spike has been illustrated [8,9,29]. Following that, we considered it to be of great importance to explore the fusion mechanism of this endemic and circulating human CoVs. In this study, firstly, we have solved the fusion core structure of HCoV-OC43 at 3.34 Å resolution, thus further adding to the structural gallery of HCoV fusion cores (Figure 5B). The formation of canonical 6-HB structure by OC43-HR1P and OC43-HR2 seems to abide by the rule accepted by other class I fusion proteins. Three HR1 helices form the central coiled coil by hydrophobic interactions among “ga” and “d” residues in HR1 domains. HR2, thereafter, binds to the hydrophobic groove formed by three HR1 chains. We noted that an extensive contact network between HR1P and HR2P, involving both hydrophilic and hydrogen bond interactions is formed. In addition, the side groove of the HR1 coiled coil is deep for its N-terminal part and relatively shallow for the C-terminal part, complementarily accommodating the helical and extended-loop halves of HR2P, respectively, which maintains the integrity of the fusion core in the post-fusion state. When comparing with the 6-HB of other common coronaviruses causing mild respiratory disease, such as 229E and NL63, the HCoV-OC43 6-HB has a similar overall structure, except for the different length of HR2 helix in the 6-HB. The HR2 domain of 229E or NL63 forms a longer loop–helix–loop structure to interact with trimeric HR1 core (Figure 5B). Overall, this finding suggested that 6-HB fusion core is a common feature among many characterized CoV fusion proteins and provides a tangible target for small molecule and peptide fusion inhibitor design by targeting HR1 substructure.

As one of fusion inhibitors targeting HR1 region, we also investigated how EK1 interacts with OC43-HR1P, in retrospect. As observed in the structure of HR1(OC43)-EK1 and HR1(OC43)-HR2, we summarized the interactions between EK1 and OC43-HR1P and those between OC43-HR2P and OC43-HR1P (Figure 4A). Most hydrophobic interactions between EK1 and OC43-HR1P are similar to those between OC43-HR2P and OC43-HR1P [compare Figure 3B and Figure 4E (left and middle panel)]. Nevertheless, EK1 compromises amounts of side chain–to–side chain plus side chain–to–main chain hydrophilic interactions with OC43-HR1P than does OC43-HR2P [compare Figure 3C (right panel) and Figure 4E (right panel)], likely due to enhanced broad-spectrum property. In sum, what we have done sheds light on understanding of the fusion mechanism of HCoV-OC43 and future design of fusion inhibitor of CoVs.

## Figures and Tables

**Figure 1 viruses-17-00343-f001:**
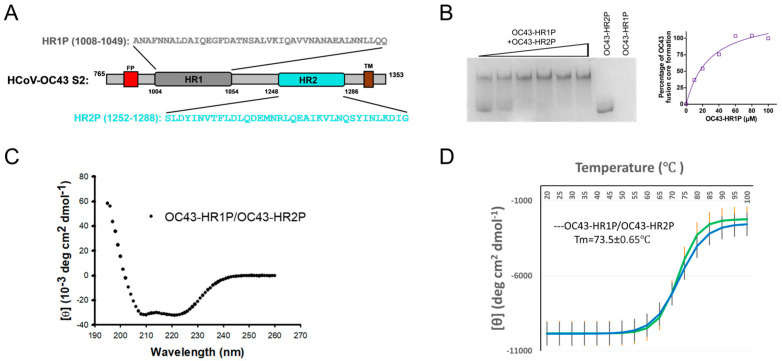
Design and biophysical characterization of OC43-HR1P and OC43-HR2P. (**A**) Schematic representation of HCoV-OC43 S protein S2 subunit, fusion peptide (FP), heptad repeat 1 (HR1), heptad repeat 2 (HR2), transmembrane domain (TM) and cytoplasmic domain (CP). OC43-HR1P and OC43-HR2P, derived from HR1 and HR2 domains, respectively, and their sequences are shown in the diagram. (**B**) Native PAGE analysis of the interaction of designed OC43-HR2P at 40 µM with OC43-HR1P at the gradient concentrations of 10, 20, 40, 60, 80 and 100 µM, respectively. The upper band indicates the six helical bundle (6-HB) formed by OC43-HR1P and OC43-HR2P, while the lower band indicates OC43-HR2P. Corresponding curve plotting the correlation between complex formation and gradient concentrations of OC43-HR1P with 40 µM OC43-HR1P shown alongside. (**C**) Circular-dichroism (CD) spectra for OC43-HR1P and OC43-HR2P complex in phosphate-buffered saline (pH 7.2). (**D**) Melting curve of the complex formed by OC43-HR1P and OC43-HR2P at 25 µM. Data (mean ± SD) were collected from two independent experiments. Signals from each temperature point were recorded three times.

**Figure 2 viruses-17-00343-f002:**
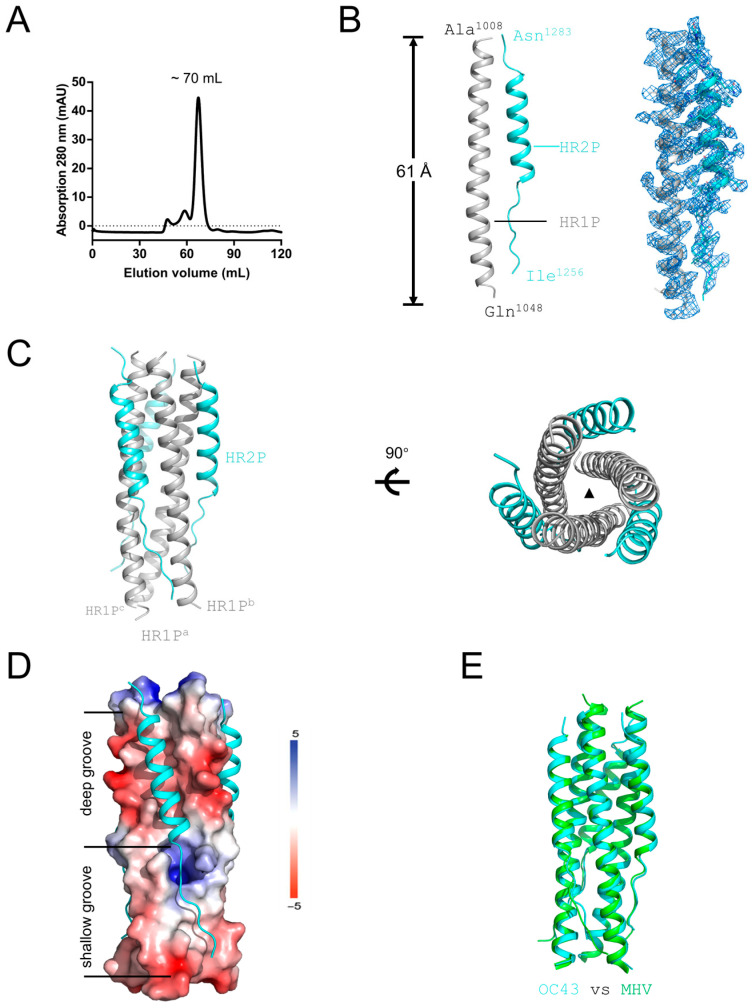
Structure of HCoV-OC43 fusion core. (**A**) SEC profile of OC43-HR1P/OC43-HR2P complex run on a Superdex 75 pg column. (**B**) Overall structure of monomeric HR1P/HR2P complex. HR1 and HR2 domains are indicated. Clear electron densities were observed for residues from A1008 to Q1048 and I1256 to N1283, respectively. These terminal residues are labeled. Quality of electron density map was exhibited alongside, correspondingly. Refined 2Fo−Fc electron density map of the peptides (contoured at 1σ and within 2 Å of selected atoms) are located between A^1008^-Q^1048^ for HR1and I^1256^-N^1283^ for HR2, respectively; (**C**) Six-helix bundle fusion core structure. The three HR1/HR2 chains are grey and cyan, respectively. The rough size of the bundle is indicated. Left side shows the side view; right side shows the top view. The three HR1 helices, labeled HR1P^a-c^, are depicted on the bottom side. (**D**) Binding of HR2 to the HR1 side groove shown as an electrostatic surface. Deep and shallow parts of the groove are indicated. (**E**) Structural comparison between OC43 (cyan) and MHV (green) fusion cores.

**Figure 3 viruses-17-00343-f003:**
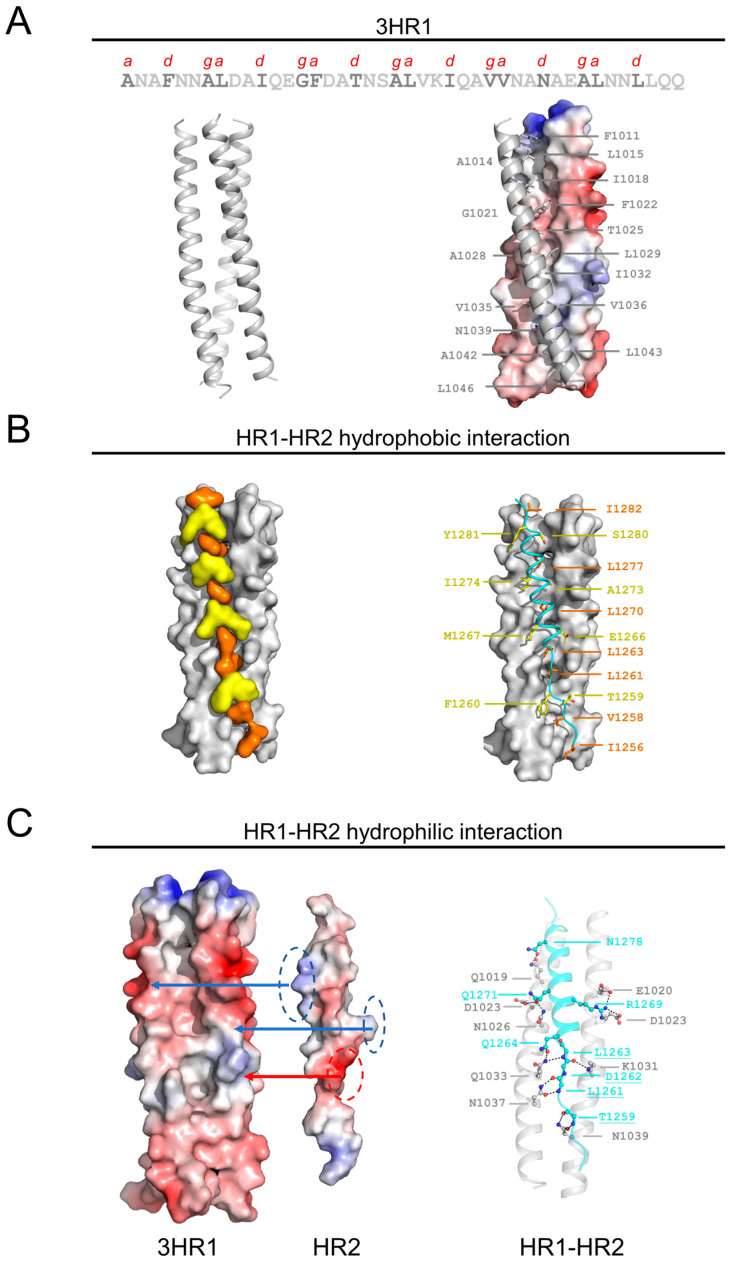
Interactions within HCoV-OC43 6-HB fusion core. (**A**) Three HR1 helices are tightly packed together in the fusion core. Letters above sequence indicate predicted hydrophobic residues at “ga” and “d” positions in HR1 regions. (**B**) Binding of HR2 to 3HR1 hydrophobic groove. Surface representation of HR1 and HR2 helices illustrates that HR2 residues fit snugly onto the surface of 3HR1core of HCoV-OC43, thereby filling HR1 hydrophobic cavities and masking its hydrophobic surface (Left panel). 3HR1 core is shown as protein surface, and HR2 residues involved in hydrophobic interactions are depicted in orange (completely buried) and yellow (packing, ~50% buried) surfaces, respectively; HR2 helices are shown as cyan ribbons on the dark grey surface of the 3HR1 cores (Right panel). HR2 residues that bury their side chains completely into the cavities on HR1 are shown as orange stick models and HR2 residues that pack around 50% of the solvent accessible surface of their side chains on ridges of HR1 are depicted as yellow stick models. (**C**) An electrostatic surface of HR1 groove is shown on the left side, while one of the HR1 helices is further presented on the right side (**Left** panel). Hydrophilic interactions for HR2 domain against HR1 trimer core (**Right** panel).

**Figure 4 viruses-17-00343-f004:**
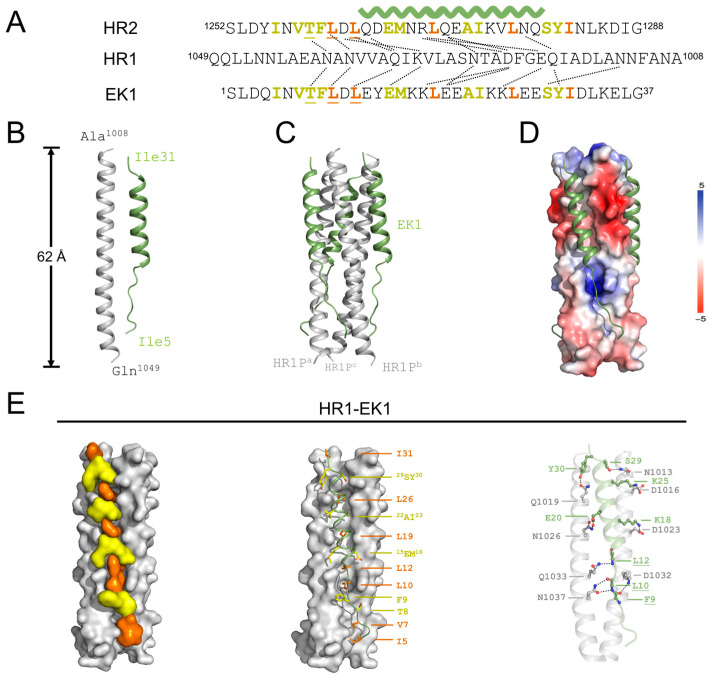
Structure of HR1(OC43)-EK1. (**A**) Design of EK1 reserving all burying and ridge-packing hydrophobic residues from OC43-HR2P. Burying residues are labelled in orange, while ridge-packing residues in light yellow. HR1 residues that mediate conserved side chain-to-side chain and side chain-to-main chain hydrophilic interactions with EK1 or OC43-HR2P residues are linked with dashed lines. The green wave symbol means helical region; monomeric (**B**) and trimeric (**C**) structures of the HR1(OC43)-EK1 complex. Six-helix bundle fusion core structure is yielded by symmetry operations. (**D**) Packing of HR2 against central hydrophobic core of HR1, as illustrated by a solvent-accessible surface rendering. Solvent-accessible surface is colored according to electrostatic potential, which ranges from +5 V (most positive, dark blue) to −5 V (most negative, dark red), with hydrophobic in white. (**E**) Interactions between OC43-HR1P and EK1.

**Figure 5 viruses-17-00343-f005:**
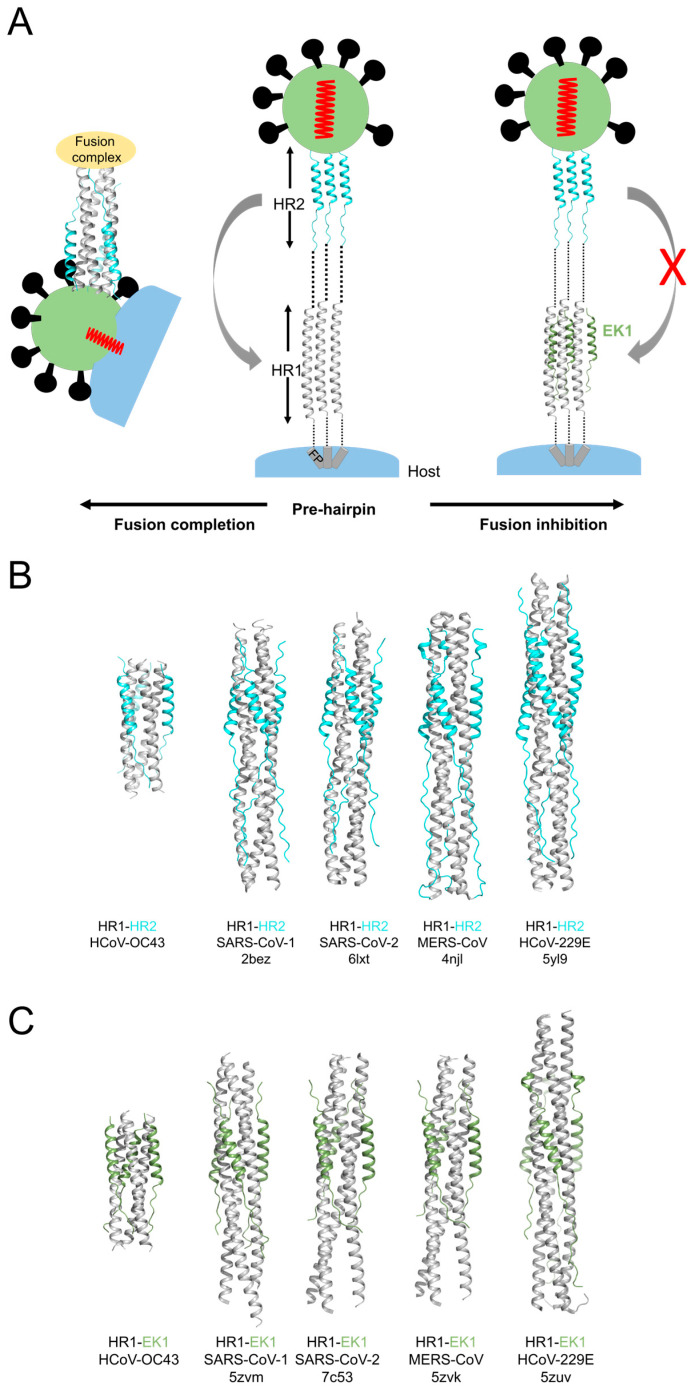
Proposed mechanism of action of fusion and fusion inhibition. (**A**) Pre-harpin conformation (middle), conformation of fusion core (**left**) and a putative intermediate state where fusion inhibitor peptides compete with HR2 in binding to trimeric HR1s (**right**). HR2 assists in fusion between viral and host cell membranes by bringing them into close proximity, then membrane fusion occurs (also corresponding to Figure 5B). Fusion inhibitor peptides bind to HR2 and prevent the recognition of HR2 onto its cognate HR1, hence inhibiting viral fusion and entry into the host cell (also corresponding to Figure 5C). Crystal gallery of fusion cores from HCoV-OC43, SARS-CoV-1 [22], SARS-CoV-2 [23], MERS-CoV [24] and HCoV-229E [25] (**B**) as well as their corresponding HR1s in complex with EK1 (**C**). Other structures of HCoV fusion core, including HCoV-NL63 and CCoV-HuPn-2018, are not shown since they lack corresponding HR1-EK1 structures [26,27].

**Table 1 viruses-17-00343-t001:** Statistics of X-ray crystallographic data processing and refinement.

	HR1(OC43)-HR2	HR1(OC43)-EK1
Data collection statistics		
Beamline	SSRF-BL19U1	SSRF-BL02U1
Wavelength (Å)	0.9789	0.9793
Space group	P222_1_	P3
Cell dimensions		
a, b, c (Å)	39.0, 48.2, 103.3	53.0, 53.0, 62.1
α, β, γ (°)	90, 90, 90	90, 90, 120
Resolution range (Å) *	48.19–3.34 (3.46–3.34)	25.74–2.71 (2.80–2.71)
No. of unique reflections	2946 (206)	5327 (548)
Completeness (%) *	94.2 (67.5)	99.4 (99.6)
<I/σ(I)> *	5.3 (2.1)	16.3 (2.3)
CC_1/2_ *^,a^	0.984 (0.359)	0.989 (0.652)
Wilson B-factor (Å^2^)	33.56	35.27
Refinement statistics		
Reflections used in refinement *	2930 (206)	5326 (548)
Reflections used for R_free_ *	134 (6)	285 (16)
R_work_ (%) *^,b^	24.0 (16.4)	28.2 (24.8)
R_free_ (%) *^,b^	28.1 (33.1)	31.1 (28.9)
No. of non-hydrogen atoms		
Protein	1564	1578
Solvent	3	/
Average B-values (Å^2^)		
Protein	27.34	49.02
Solvent	30	/
RMSD bond length (Å)	0.01	0.005
RMSD bond angle (°)	1.38	0.78
Ramachandran favored (%)	94.76	99.48
Ramachandran allowed (%)	4.72	0.52
Ramachandran outlier (%)	0.52	0
PDB entry	8ZX7	9KMO

* Statistics for the highest-resolution shell are shown in parentheses. ^a^
${CC}_{\frac{1}{2}}=\frac{\sum (x-\langle x\rangle )(y-\langle y\rangle )}{{\left[\sum {\left(x-\langle x\rangle \right)}^{2}\sum {\left(y-\langle y\rangle \right)}^{2}\right]}^{\frac{1}{2}}}$; ^b^ R_work_ $=\frac{\sum \left|\left|F_{obs}\right|-\left|F_{calc}\right|\right|}{\sum \left|F_{obs}\right|}$; R_free_ is defined as R_work_ calculated from 5% of the reflections that were excluded from refinement.

## Data Availability

Structures of HR1(OC43)-HR2 and HR1(OC43)-EK1 were deposited in the RCSB PDB database (RCSB PDB: Homepage (https://www.rcsb.org/)) with the entries of 8ZX7 and 9KMO, respectively.

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
