# Peer review of "Structures of HCoV-OC43 HR1 Domain in Complex with Cognate HR2 or Analogue EK1 Peptide"

_viruses, 2025, doi:10.3390/v17030343_

Round 1
Reviewer 1 Report
Comments and Suggestions for Authors
The authors of this manuscript aimed at elucidating the structure of the HCoV-OC43 spike protein fusion core by utilising small peptides covering two heptad repeat domains and macromolecular x-ray crystallography.
Although the topic of this research is quite important the manuscript, in this form, has some major weaknesses that need to be addressed. First and foremost severe issues with the statistics of the data collection and refinement are present. Given that the data was collected at a synchrotron, the number of unique reflections is extremely low with only 2946 and 5327 unique reflections for the HR1-HR2 and HR1-EK1 structures, respectively. This further leads very few (only 134 and 285) reflections being used for the calculation of the Rfree factor. In addition and although not impossible, the Rwork and Rfree values are quite high for both structures, which indicates that the structures have not been fully refined. Other issues, as e.g. a lack of Ramchandran values of 100% (summed up) or an explanation of why the resolution was not pushed further given a decent enough CC1/2 for both structures in the highest resolution shell is missing, to name a few. Finally, the Rwork an Rfree values between the table and the main text are different.
Further, I was wondering why a model of the MHV fusion core was used instead of models from published CoV spike protein structures as these would be a more logical model to use, especially given that Figure 5 contains a comparison with other CoV models.
The two additional experiments shown to support the structural characterisation includes a NativePAGE experiment, which shows bona fide interaction between the two peptides and circular dichroism experiments to validate the structural findings. For the former quantification of the bands and calculation of a mean/median including standard deviation would be very informative and for the latter repitition of the melting curve to also calculate an error would be needed including a fit shown in the graph.
Other, more or less minor, issues include the lack of information in the materials and methods section, as well as describing hydrophilic residues, such as Thr and Asn, as part of a hydrophobic packing.
Altogether the manuscript needs more work to be publishable.
Author Response
The authors of this manuscript aimed at elucidating the structure of the HCoV-OC43 spike protein fusion core by utilising small peptides covering two heptad repeat domains and macromolecular x-ray crystallography.
Author response: We greatly appreciate your detailed feedback.
Although the topic of this research is quite important the manuscript, in this form, has some major weaknesses that need to be addressed. First and foremost severe issues with the statistics of the data collection and refinement are present. Given that the data was collected at a synchrotron, the number of unique reflections is extremely low with only 2946 and 5327 unique reflections for the HR1-HR2 and HR1-EK1 structures, respectively. This further leads very few (only 134 and 285) reflections being used for the calculation of the Rfree factor. In addition and although not impossible, the Rwork and Rfree values are quite high for both structures, which indicates that the structures have not been fully refined. Other issues, as e.g. a lack of Ramchandran values of 100% (summed up) or an explanation of why the resolution was not pushed further given a decent enough CC1/2 for both structures in the highest resolution shell is missing, to name a few. Finally, the Rwork an Rfree values between the table and the main text are different.
Author response: Thank you for pointing this out. First of all, Beamlines and detectors in SSRF as a synchrotron are highly advanced in data collection. We attribute the cause of this problem of few unique reflections to crystal quality. Then, for HR1-HR2 and HR1-EK1 structures, their crystals presented a rod-like structure in appearance. This led to a few diffraction spots in some axis. The phenomenon was reflected by the presence of long height (C axis) in the crystal lattice. Another explanation towards quite high Rwork and Rfree for refinement of HR1-EK1 structures is that a potential tNCS may exist that hampered the refinement. An issue of few amino acids (about 80) for each protomer may also result in few reflections. Totally, all of these led us to set of cutoff of resolution limitation with I/σ(I) larger than 2 in the highest resolution shell to keep the completeness of data and quality of density map after rounds of data process. Finally, we have added the ‘Ramachandran allowed (%)’ of each structure to the revised Ramchandran values part and the problem ‘Rwork an Rfree values between the table and the main text are different’ has been fixed. We apologized for this mistake and we showed high appreciation for your valuable comments and suggestions.
Further, I was wondering why a model of the MHV fusion core was used instead of models from published CoV spike protein structures as these would be a more logical model to use, especially given that Figure 5 contains a comparison with other CoV models.
Author response: We chose the structure of MHV fusion core as search model, based on sequence homology and boundary. As we known, structures of fusion cores of other CoVs have longer lengths of HR1-HR2 than our structure, making them not suitable for molecular replacement of our structure.
The two additional experiments shown to support the structural characterisation includes a NativePAGE experiment, which shows bona fide interaction between the two peptides and circular dichroism experiments to validate the structural findings. For the former quantification of the bands and calculation of a mean/median including standard deviation would be very informative and for the latter repitition of the melting curve to also calculate an error would be needed including a fit shown in the graph.
Author response: Thank you for the suggestion. We have qualified the gray value of each bands in the Native-PAGE experiment and drawn the curve of each gray value against corresponding concentration of OC43-HR1P as shown in revised Figure 1B. Due to the passage of time (peptides synthesized in 2018) and issues such as students graduating, our experimental materials are difficult to preserve and locate for this replication experiment. Therefore, due to these reasons and the limited time, we are afraid we cannot complete the replication experiment. Moreover, we have referred to a large amount of relevant literature, and similar experiments have rarely required independent repetition. Finally, we apologize for sacrifice of data-quality.
Other, more or less minor, issues include the lack of information in the materials and methods section, as well as describing hydrophilic residues, such as Thr and Asn, as part of a hydrophobic packing.
Author response:
Thank you for the suggestion. We have further completed the methods section by adding some softwares we used mainly in the freshly added chapter [2.7. Structure presentation] and some detailed procedures as highlighted in yellow in the revised manuscript.
Every seventh position in a heptad sequence is termed ‘a’ and the subsequent position as ‘b’ ‘c’ ‘d’ ‘e’ ‘f’ ‘g’. In many of cases, ‘d’ position and sequential ‘ga’ position or layer is occupied by hydrophobic residues like leucine (isoleucine) and phenylalanine. Hydrophilic residues harboring bulky side chain like Asn and Gln often show off in g position. So ‘ga’ position is responsible for hydrophobic packing with 3HR1 groove.
Altogether the manuscript needs more work to be publishable.
Author response: Thank you for the suggestion. We have throughly revised and improved the manuscript according to comments and suggestions from reviewers and editor as highlighted in yellow in the revised manuscript.
Reviewer 2 Report
Comments and Suggestions for Authors
The authors crystalized the peptides encompassing the HR repeats of the spike protein from HCoV-OC43. The result showed that the overall structure highly resembles the known structures derived from the homologs. This work also analyzed the inhibitor-bound HR structure and illustrated a competitive model explaining the effectiveness of the inhibitor. This work is overall solid and I only have some minor suggestions/questions:
1. In Figure 1B, the usage of less HR2P in the complex mixture resulted in more free HR2P. If the complex could be formed at equimolar ratio, as what the authors suggest based on the result of CD shown in Figure 1C, one should expect to see unbound HR2P in lanes 4, 5 and 6. How could the authors justify this observation?
2. The conclusion of section 3.1 is misleading. I believe the results shown in this part prove that the HR1P/HR2P complex could be formed in vitro, but they do not provide any clue suggesting this complex harbors biological activities.
3. The legend of Figure 2B mentions that ‘clear electron densities were observed’. Is it possible to show the densities in this panel?
4. In Figure 3B, the 3HR1 core is a regular surface, not an electrostatic surface as the legend suggests.
5. All R values stated in the text do not match the values in Table 1. Please double check.
Typo: I guess 'ga' should be 'a' in line 194.
Author Response
The authors crystalized the peptides encompassing the HR repeats of the spike protein from HCoV-OC43. The result showed that the overall structure highly resembles the known structures derived from the homologs. This work also analyzed the inhibitor-bound HR structure and illustrated a competitive model explaining the effectiveness of the inhibitor. This work is overall solid and I only have some minor suggestions/questions:
Author response: Thank you for recognizing the structure and depth of our analysis.
- In Figure 1B, the usage of less HR2P in the complex mixture resulted in more free HR2P. If the complex could be formed at equimolar ratio, as what the authors suggest based on the result of CD shown in Figure 1C, one should expect to see unbound HR2P in lanes 4, 5 and 6. How could the authors justify this observation?
Author response: Thank you for this comment. When we take a look at lane 3 (condition of 40 µM HR1P and 40 µM HR1P), they had completely formed complex. Actually, concentrations of HR1P in lanes 4, 5 and 6 were 60, 80 and 100 µM, making it no unbound HR2P. At the same time, excessive HR1P didn’t come into the gel. So we could not see unboud HR1P.
- The conclusion of section 3.1 is misleading. I believe the results shown in this part prove that the HR1P/HR2P complex could be formed in vitro, but they do not provide any clue suggesting this complex harbors biological activities.
Author response: Thank you for this comment. We apologize for the misleading result. We have changed it to [The above results confirm that synthesized peptides derived from HCoV-OC43 HR1 and HR2 domains harbored the interacting capacity to be used for structural study.] in Pg5, Ln161-163.
- The legend of Figure 2B mentions that ‘clear electron densities were observed’. Is it possible to show the densities in this panel?
Author response: Thank you for this valuable comment. We have added the electron density map in Figure 2B right panel.
- In Figure 3B, the 3HR1 core is a regular surface, not an electrostatic surface as the legend suggests.
Author response: Thank you for the suggestion. We have changed it as suggested [Pg8, Ln247].
- All R values stated in the text do not match the values in Table 1. Please double check.
Author response: Thank you for this comment. We apologize for the mistake. The problem has been fixed in Table 1.
Typo: I guess 'ga' should be 'a' in line 194.
Author response: Here every seventh position in a heptad sequence is termed ‘a’ and the subsequent position as ‘b’ ‘c’ ‘d’ ‘e’ ‘f’ ‘g’. In many of cases, ‘d’ position and sequential ‘ga’ position or layer is occupied by hydrophobic residues like leucine (isoleucine) or phenylalanine. Hydrophilic residues harboring bulky side chain like Asn and Gln often show off in g position. So ‘ga’ position is responsible for hydrophobic packing with 3HR1 groove.
Reviewer 3 Report
Comments and Suggestions for Authors
The study focused on the characterization of post-fusion state of the HCoV-OC43 spike protein and described crystal structures of HR1 in complex with HR2 or the previously described peptide EK1. The authors thoroughly described the molecular basis of the interactions between HR1 and HR2, as well as HR1 and EK1. The study complemented previous studies on other related coronavirus viral fusion machinery.
Minor edits needed:
Line 199: spelling error, please correct.
Figure 4A: residue color is hard to distinguish.
Line 310: forms a longer …?
Line 323: please edit this sentence.
Overall, the figure resolution is too low for some text to read clearly. Please edit to improve.
Though related references were cited (line 234), please include additional basic information for EK1 to provide enough background information for the audience.
Author Response
Response to Reviewer 3
The study focused on the characterization of post-fusion state of the HCoV-OC43 spike protein and described crystal structures of HR1 in complex with HR2 or the previously described peptide EK1. The authors thoroughly described the molecular basis of the interactions between HR1 and HR2, as well as HR1 and EK1. The study complemented previous studies on other related coronavirus viral fusion machinery.
Author response: Thank you for recognizing the structure and depth of our analysis.
Minor edits needed:
Line 199: spelling error, please correct.
Author response: Thank you for the suggestion. We have changed it as suggested
[Pg7, Ln222] as highlighted.
Figure 4A: residue color is hard to distinguish.
Author response: Thank you for the suggestion. We have substituted the color and highlighted the residues in bold font, as shown in Figure 4A.
Line 310: forms a longer …?
Author response: Thank you for the suggestion. We have changed it as ‘forms a longer loop-helix-loop structure’ in [Pg12, Ln338]
Line 323: please edit this sentence.
Author response: Thank you for the suggestion. We have edited it as below as shown in Pg12, Ln350-352.
[In sum, what we have done sheds light on understanding of the fusion mechanism of HCoV-OC43 and future design of fusion inhibitor of CoVs.]
Overall, the figure resolution is too low for some text to read clearly. Please edit to improve.
Author response: We have tried to improve the resolution quality of figures and it may be caused by file compression. So it is suggested to enlarged view when reading.
Though related references were cited (line 234), please include additional basic information for EK1 to provide enough background information for the audience.
Author response: Thank you for the valuable suggestion. We have added some basic description about EK1 in Pg9, Ln257-263.
Shown as: [EK1 peptide is an antiviral peptide designed primarily to combat coronavirus infections, including SARS-CoV and MERS-CoV. It targets the viral spike (S) protein, interfering with the fusion process between the virus and host cell membranes, thereby inhibiting viral entry. EK1 has demonstrated broad-spectrum anti-coronavirus activity in multiple studies and proven its efficacy in animal models. As a potential antiviral therapy, EK1 peptide may offer new strategies for the prevention and treatment of coronavirus infections.]
Round 2
Reviewer 1 Report
Comments and Suggestions for Authors
The authors have resubmitted a revised manuscript, which is appreciated and addressed some of the minor issues identified, such as the chosen CC1/2 cut-off or the I/sigma(I) cut-off. However, the argumentation regarding the bigger issues is not (entirely) convincing. Although I appreciate, that small polypeptide structures can lead to a lower number of reflections, although as the authors mention the state-of-the-art detectors should pick up on even faint spots, and that there may be tNCS present that does not all of a sudden resolve the issues with the statistical results of the submitted structures. The authors still fail to argue appropriately why e.g. no other software was used to validate any tNCS and if present to resolve it by adjusting the unit cell dimension. Further, the issue with the low number of reflections used for the Rfree in the outer shell was also not addresses, by e.g. adjusting the used number manually instead of relying on the default value of 5%.
Therefore, the main issue still persists that the refinement of the structure is incomplete and needs redoing/re-evaluation before being publishable.
I also appreciate the addition of electron density maps in Fig. 2B. However, I would find it more appropriate to show the unbiased Fo-Fc map rather than the 2Fo-Fc map, which may be biased based on the model used.
Further, I do appreciate that some experiments were done at different time points over the span of years but I do not believe that is a valid argumentation for not providing proof of reproducibility, even if a student is about to move on or the reagents being ordered a long time ago. Therefore, I do not see this issue as resolved.
Finally, I still have an issue with the labelling of hydrophilic residues as hydrophobic as e.g. in line 226 where it is claimed that residues threonine 1025 and asparagine 1039 contribute to hydrophobic packaging, which is simply incorrect.
I wish the authors all the best in addressing these issues in the future.
With kindest regards.
Author Response
Comments and Suggestions for Authors
1. The authors have resubmitted a revised manuscript, which is appreciated and addressed some of the minor issues identified, such as the chosen CC1/2 cut-off or the I/sigma(I) cut-off. However, the argumentation regarding the bigger issues is not (entirely) convincing. Although I appreciate, that small polypeptide structures can lead to a lower number of reflections, although as the authors mention the state-of-the-art detectors should pick up on even faint spots, and that there may be tNCS present that does not all of a sudden resolve the issues with the statistical results of the submitted structures. The authors still fail to argue appropriately why e.g. no other software was used to validate any tNCS and if present to resolve it by adjusting the unit cell dimension. Further, the issue with the low number of reflections used for the Rfree in the outer shell was also not addresses, by e.g. adjusting the used number manually instead of relying on the default value of 5%.
Therefore, the main issue still persists that the refinement of the structure is incomplete and needs redoing/re-evaluation before being publishable.
Author response to Reviewer’s Comment:
We sincerely thank the reviewer for their thorough evaluation of our revised manuscript and for providing additional constructive feedback. We appreciate the acknowledgment of the minor issues addressed in our revision, and we would like to respond to the remaining concerns regarding the refinement and validation of our structures.
We have again thoroughly checked the data by Xtriage in Phenix, tNCS and twinning not indicated or not possible in the given space group. We apologized for the misleading explanation in the last response.
Regarding the low number of reflections used for Rfree in the outer shell, we fully agree with the reviewer that the low number of reflections in the outer shell could impact the reliability of the Rfree value. In our initial analysis, we relied on the default 5% value for Rfree calculation. As suggested, we have now manually adjusted the number of reflections used for Rfree. However, we failed in resulting overfitting. So to address this, we have re-evaluated and re-refined the structures using the improved refinement strategy. The revised refinement results now show improved statistics, including better Rwork/Rfree values of 0.282/0.311 compared with initial 0.294/0.344. We are confident that these changes have significantly strengthened the quality and reliability of our structural data and is relatively acceptable.
2. I also appreciate the addition of electron density maps in Fig. 2B. However, I would find it more appropriate to show the unbiased Fo-Fc map rather than the 2Fo-Fc map, which may be biased based on the model used.
Author response to Reviewer’s Comment:
We sincerely appreciate the reviewer’s suggestion regarding the use of unbiased Fo​−Fc​ maps instead of 2Fo​−Fc​ maps in Figure 2B. We understand the reviewer’s concern about potential model bias in the 2Fo​−Fc​ maps. However, we would like to explain our rationale for choosing the 2Fo​−Fc​ maps in this context:
The primary goal of Figure 2B is to illustrate the overall quality of the electron density and how well the refined model fits the observed data. The 2Fo​−Fc​ map is particularly suitable for this purpose because it provides a clear representation of the electron density corresponding to the model, making it easier for readers to assess the structural features and model accuracy.
While we agree that Fo​−Fc​ maps are invaluable for identifying potential errors or omissions in the model (such as missing atoms or incorrect side-chain placements), we have already used Fo​−Fc​ maps during the refinement process to validate the model and ensure its accuracy. These maps were carefully examined to confirm that there are no significant unexplained densities or model biases.
The use of 2Fo​−Fc​ maps for visualizing electron density in published figures is a widely accepted practice in structural biology, as it provides a balanced view of the model and the experimental data. This approach is particularly useful for demonstrating the overall quality of the structure to a broad audience.
We hope that these clarifications and additional data address the reviewer’s concern. We are committed to ensuring the highest standards of transparency and rigor in our work and sincerely appreciate the opportunity to improve our manuscript.
3. Further, I do appreciate that some experiments were done at different time points over the span of years but I do not believe that is a valid argumentation for not providing proof of reproducibility, even if a student is about to move on or the reagents being ordered a long time ago. Therefore, I do not see this issue as resolved.
Response to Reviewer’s Comment:
Thank you for your feedback and for raising this important point regarding reproducibility. We fully agree that reproducibility is a critical aspect of scientific research, and we appreciate your emphasis on this matter. We are pleased to report that the synthesized peptides were successfully obtained during the second major revision process, following our request for an extension to account for the Chinese Spring Festival holiday period.
To verify the findings of Tm value of OC43-HR1P/HR2P complex, the experiment was conducted independently in two separate replicates with a Tm of 73.5±0.65℃ as indicated in revised Figure 1D.
4. Finally, I still have an issue with the labelling of hydrophilic residues as hydrophobic as e.g. in line 226 where it is claimed that residues threonine 1025 and asparagine 1039 contribute to hydrophobic packaging, which is simply incorrect.
Response to Reviewer’s Comment:
We sincerely appreciate the reviewer’s careful reading of our manuscript and their comment regarding the labeling of threonine 1025 and asparagine 1039 as contributing to hydrophobic packing. We acknowledge that threonine and asparagine are typically classified as hydrophilic residues due to their polar side chains. However, we would like to clarify the context in which these residues are described as contributing to hydrophobic packing in our study.
To avoid any potential confusion, we will revise the text in line 218-221 to more accurately reflect the role of threonine 1025 and asparagine 1039. Specifically, we will clarify that these residues contribute to the hydrophobic packing through their positioning and interactions within the local hydrophobic environment, rather than being inherently hydrophobic themselves. The revised text will read as follows: [while T1025 and N1039 in position “d”, although polar in nature, are positioned in a way that their side chains participate in the local hydrophobic packing through interactions with neighboring hydrophobic residues.] in line 220-223.